# *Neospora caninum*: Structure and Fate of Multinucleated Complexes Induced by the Bumped Kinase Inhibitor BKI-1294

**DOI:** 10.3390/pathogens9050382

**Published:** 2020-05-16

**Authors:** Pablo Winzer, Nicoleta Anghel, Dennis Imhof, Vreni Balmer, Luis-Miguel Ortega-Mora, Kayode K. Ojo, Wesley C. Van Voorhis, Joachim Müller, Andrew Hemphill

**Affiliations:** 1Institute of Parasitology, Vetsuisse Faculty, University of Bern, 3012 Bern, Switzerland; pablo.winzer@vetsuisse.unibe.ch (P.W.); nicoleta.anghel@vetsuisse.unibe.ch (N.A.); dennis.imhof@vetsuisse.unibe.ch (D.I.); vreni.balmer@vetsuisse.unibe.ch (V.B.); joachim.mueller@vetsuisse.unibe.ch (J.M.); 2Graduate School for Cellular and Biomedical Sciences, University of Bern, 3012 Bern, Switzerland; 3SALUVET, Animal Health Department, Faculty of Veterinary Sciences, Complutense University of Madrid, Ciudad Universitaria, 28040 Madrid, Spain; luisucm@ucm.es; 4Center for Emerging and Re-emerging Infectious Diseases (CERID), Division of Allergy and Infectious Diseases, Department of Medicine, University of Washington, Seattle, WA 98109, USA; ojo67kk@yahoo.ca (K.K.O.); wesley@uw.edu (W.C.V.V.)

**Keywords:** bumped kinase inhibitor, calcium-dependent protein kinase, immunofluorescence, immunogold labeling, inner membrane complex, neosporosis, surface antigen, tachyzoite, transmission electron microscopy

## Abstract

Background: Bumped kinase inhibitors (BKIs) are potential drugs for neosporosis treatment in farm animals. BKI-1294 exposure results in the formation of multinucleated complexes (MNCs), which remain viable in vitro under constant drug pressure. We investigated the formation of BKI-1294 induced MNCs, the re-emergence of viable tachyzoites following drug removal, and the localization of CDPK1, the molecular target of BKIs. Methods: *N. caninum* tachyzoites and MNCs were studied by TEM and immunofluorescence using antibodies directed against CDPK1, and against NcSAG1 and IMC1 as markers for tachyzoites and newly formed zoites, respectively. Results: After six days of drug exposure, MNCs lacked SAG1 surface expression but remained intracellular, and formed numerous zoites incapable of disjoining from each other. Following drug removal, proliferation continued, and zoites lacking NcSAG1 emerged from the periphery of these complexes, forming infective tachyzoites after 10 days. In intracellular tachyzoites, CDPK1 was evenly distributed but shifted towards the apical part once parasites were extracellular. This shift was not affected by BKI-1294. Conclusions: CDPK1 has a dynamic distribution depending on whether parasites are located within a host cell or outside. During MNC-to-tachyzoite reconversion newly formed tachyzoites are generated directly from MNCs through zoites of unknown surface antigen composition. Further in vivo studies are needed to determine if MNCs could lead to a persistent reservoir of infection after BKI treatment.

## 1. Introduction

The phylum Apicomplexa includes important pathogens of animals and/or man such as *Plasmodium falciparum, Cryptosporidium parvum, Eimeria* spp., *Babesi spp, Theileria spp., Toxoplasma gondii,* and *Neospora caninum* among others. *N. caninum* is closely related to *T. gondii*, can infect many different intermediate hosts and is able to proliferate in a wide range of tissues and in many different cell types in vitro. However, in contrast to *T. gondii*, *N. caninum* is an important causative agent of abortion or birth of weak offspring in cattle, and to a lesser extent, in sheep and other ruminants [1]. The sexual cycle of *N. caninum* occurs in dogs, which do not only shed oocysts that become infective after sporulation but can also act as intermediate host and, are affected by neurological clinical signs. No vaccine is currently licensed for the prevention of bovine or canine neosporosis and so far, immuno- or chemotherapeutical treatments do not exist [2,3].

Protein kinases are involved in a plethora of different functional activities that control essential aspects of apicomplexan biology, including host cell invasion, intracellular proliferation and egress [4,5]. Calcium-dependent protein kinases (CDPKs) are not found in mammals, and therefore constitute interesting targets for anti-apicomplexan drugs. In particular, CDPK1 is intensively studied in target-based drug development against a wide range of apicomplexans including *P. falciparum*, *T. gondii*, *N. caninum*, *Sarcocystis neurona*, *Besnoitia besnoiti*, *B. bovis*, *T. equi* and *C. parvum* [6]. 

Based on co-crystal structure investigations of *T. gondii* CDPK1, a class of ATP-competitive inhibitors named bumped kinase inhibitors (BKIs) has been developed [7]. They exhibit a high degree of efficacy and specificity for apicomplexan CDPK1 relative to mammalian kinases, and they block the invasion of *T. gondii* tachyzoites into host cells [8] and egress [9] from host cells. These BKIs fit into the ATP binding cavity not only of *T. gondii* CDPK1 but also of CDPK1 isoforms of other apicomplexans including *N. caninum* [10].

Consequently, several BKIs have been studied so far with respect to efficacy against *N. caninum* infection, in particular the pyrazolopyrimidine BKI-1294. BKI-1294 interferes with host cell invasion and egress but is not parasiticidal [10]. BKI-1294 is highly active in pregnant mouse models for neosporosis [11] and for toxoplasmosis [12], and in a pregnant sheep model for toxoplasmosis [13]. Other BKIs are also effective against neosporosis, as shown in pregnant mouse [14] and sheep [15] models. 

It is unclear, so far, how BKI-1294 (and related compounds) affect the intracellular stages of apicomplexans. We have demonstrated in previous studies that prolonged exposure of infected cells to this compound results in the formation of multinucleated complexes (MNCs) [10]. We have reproduced and confirmed this observation with other *N. caninum* strains (Nc-1, Nc-Liv, and Nc-Spain7), and with the *T. gondii* strains RH and ME49. It was further observed that BKI-1294 induced MNCs exhibit a deregulated gene expression pattern as evidenced by the simultaneous expression of bradyzoite and tachyzoite antigens [11]. These results suggest that MNCs could constitute a drug-induced resting stage assuring the survival of the parasite until the release of the drug pressure.

To get an insight into the cell biology of MNCs, we have investigated their ultrastructure, the spatiotemporal pattern of established marker proteins for *N. caninum* development, and the pattern of NcCDPK1, the BKI target, in MNCs in comparison to intra- and extracellular tachyzoites. Moreover, we show how newly formed tachyzoites are generated once the drug is removed and try to answer the question of whether tachyzoite reconversion is achieved by only a few drug-resistant/drug-adapted parasites, or whether newly formed tachyzoites are generated directly from MNCs.

## 2. Results

### 2.1. Ultrastructural Characteristics of Multinucleated Complexes (MNCs)

Transmission electron microscopy (TEM) micrographs of *N. caninum* tachyzoites grown in human foreskin fibroblasts (HFF) monolayers for two days in the absence of BKI-1294 are shown in Figure 1A, and a higher magnification view is shown in Figure 1B. After two days of culture, intracellularly proliferating tachyzoites were located within a parasitophorous vacuole (PV), which is surrounded by a parasitophorous vacuole membrane (PVM) (Figure 1A). The PV is in close proximity to the host cell nucleus, and often associated with host cell mitochondria (Figure 1B). Dense granules, rhoptries, and micronemes are hallmarks of apicomplexan parasites and could be easily identified. Tachyzoites were embedded in the PV-matrix, which consists of a network of membranous tubules. Individual tachyzoites were always surrounded by a thick plasma membrane that contains three layers, namely an inner membrane complex (IMC), and a double-layered plasmalemma at the outer surface. 

After two days of culture in the presence of 5 µM BKI-1294, small MNCs containing 3–5 nuclei were found. An example is shown in Figure 2A and at lower magnification in Figure 2B. MNCs were also located within a PV, and exhibited similar features as the individual tachyzoites, including an intact conoid, and secretory organelles (dense granules, rhoptries, and micronemes). Higher magnification shows that MNCs were surrounded by the typical three-layered plasma membrane seen in tachyzoites. However, individual zoites were formed and are separated from each other by only one membrane layer, which appears to branch off from the outer membrane (see Figure 2A, large horizontal red arrow), and continues along the surface of the newly formed zoites (small arrows in Figure 2A). In addition, the presence of an intact mitochondrion indicates that these MNCs were viable and metabolically active. The matrix of the PV contained similar membranous tubules as in untreated cultures, but also more granular components were present.

On day four of BKI-1294 treatment (Figure 2C,D), MNCs had increased in size, as had the number of individual nuclei. Newly formed zoites with intact organelles were still visible and surrounded by a single membrane layer (red arrows in Figure 2D). On day nine of BKI-1294 treatment (Figure 3A–C), MNCs exhibited a further increase in size, zoites were densely packed, separated by a single membrane, and a larger number of newly formed conoids were evident. These MNCs were still surrounded by a triple-layered plasma membrane, and the matrix of the PV appeared more granular, while the tubulo-vesicular membranous network had largely disappeared (Figure 3C,D).

### 2.2. Expression of NcSAG1 and NcIMC1 during the Formation of MNCs and Subsequent Drug Release

Antibodies against the inner membrane complex protein 1 (IMC1) serve as markers for the IMC that surrounds newly formed zoites, while antibodies directed against the major surface antigen 1 (NcSAG1) bind exclusively to the outer layer of the tachyzoite plasma membrane. As expected, non-treated control cultures showed tachyzoites undergoing extensive proliferation, forming vacuoles containing numerous parasites that exhibited both anti-IMC1 and anti-NcSAG1 antibody labeling closely associated with their surface (Figure 4A,C,E). After four days of culture without drug treatment, host cell lysis started to occur (not shown). In cultures treated with BKI-1294 for one, two and, four days (Figure 4B,D,F), staining for IMC1 differed from NcSAG1 labeling. NcSAG1 was localized largely on the surface or peripheral regions of the MNCs, while the interior was only diffusely labeled. In addition, granular deposits were frequently seen within the host cell cytoplasm from day two of treatment onwards. In contrast, IMC1 staining was found to be associated with the plasma membrane of individual newly formed zoites in the interior of the MNCs. On day six of BKI-1294 treatment (Figure 4G), NcSAG1 labeling at the periphery of MNCs was diminished, and the corresponding staining within the complex appeared indistinct, while anti-IMC1 labeling revealed the presence of numerous newly formed zoites that were mostly devoid of anti-NcSAG1 labeling.

In the next step, we investigated the re-differentiation of MNCs to tachyzoites after the removal of BKI-1294. NcSAG1 was found on the surface of MNCs until day two post-drug removal (pdr) (Figure 5A,B). However, NcSAG1 labeling became less defined with time, and was also present as a diffuse staining located in the neighboring area of the MNCs, most likely in the host cell cytoplasm (Figure 5A,B). In contrast, staining with anti-NcIMC1 antibody marking the presence of newly formed zoites increased in the periphery (Figure 5B, green arrows). Specimens viewed on day seven lacked defined NcSAG1 surface staining, and NcSAG1 became more prominent in the MNC interior (Figure 5C). On day seven, the numbers of newly formed zoites increased with time, as did the number of zoites present at the periphery by becoming more frequent, and this was more prominently seen on day 10 pdr (Figure 5D, left panel). However, also from day 10 pdr onwards, newly formed and obviously proliferating tachyzoites could be detected within new PVs, indicating that these zoites had emerged from their MNCs, infected neighboring cells, and had converted into tachyzoites (Figure 5D, right panel). 

### 2.3. Expression of NcCDPK1 during the Formation of MNCs, Tachyzoite Egress, and Extracellular Maintenance of Tachyzoites

NcCDPK1 is the primary molecular target of BKI-1294 [10]. To investigate the distribution of NcCDPK1 during MNC formation, we employed an antibody directed against the homologous *T. gondii* CDPK1. This antibody had been previously shown to cross-react with *Sarcocystis neurona* CDPK1 [16]. On Western blots of *N. caninum* tachyzoite extracts (Figure 6A), it reacted with a 57 kDa band, which corresponded to the predicted molecular weight of NcCDPK1 [10]. This band was detected in the Triton-X-100 soluble fraction only. IF labeling of intracellular *N. caninum* tachyzoites grown in HFF resulted in evenly distributed cytoplasmic staining of parasites, but not within the parasite nuclei (Figure 6B). On day three of treatment, NcCDPK1 staining of the MNC interior was ill-defined, and not confined to individual newly formed zoites, but rather evenly distributed within the interior of the complexes (Figure 6D). On day six of treatment, the intensity of NcCDPK1 staining was reduced (Figure 6E).

### 2.4. Expression of NcCDPK1 during the Formation of BKI-1294 Induced MNCs, Egress, and Extracellular Maintenance

In tachyzoites purified at 4 °C, NcCDPK1 staining was uniformly distributed within the parasite cytoplasm (Figure 7A), similar to what was previously found in intracellular tachyzoites in situ (Figure 7B). When tachyzoites were purified at 21 °C, NcCDPK1 was located to the apical region of the tachyzoites (Figure 7B). This apical location of NcCDPK1 was also found in parasites that were exposed to 5 µM BKI-1294 for 15 min prior to separation from host cells in the presence of BKI-1294 (Figure 7C). To confirm this, infected cells were pre-fixed in paraformaldehyde for 10 min and were then subjected to host cell lysis and parasite isolation. In these cells, NcCDPK1 was distributed evenly in the cytoplasm, as seen in intracellular parasites (Figure 7D). 

When tachyzoite egress was induced by dithiothreitol (DTT), a well-established trigger of egress of tachyzoites from host cells [17], a similar apical shift in NcCDPK1 localization was observed (Figure 8A). Double IF labeling of extracellular tachyzoites with anti-CDPK1 and anti-tubulin (Figure 8B) or anti-NcMIC2 antibodies (Figure 8C) showed that the polarized NcCDPK1 staining was localized at the apical part of *N. caninum* tachyzoites. In comparative Western blots of secreted fractions versus intracellular fractions, only small amounts of NcCDPK1 were found to be translocated to the extracellular medium, indicating that only a small portion of this protein was actually secreted or released by these parasites (Figure 8D).

Immunogold-TEM of intracellular and extracellular tachyzoites was carried out in order to determine which tachyzoite compartments and/or organelles would be labeled with anti-CDPK1 antibodies. Intracellularly proliferating tachyzoites (Figure 9) exhibited a marked staining of the anterior end, mostly situated around the sub-membranous apical part of the parasites, just below the plasma membrane. Few gold particles were associated with micronemes. In addition, no or only a little immunogold-labeling could be found within the nucleus or secretory organelles such as rhoptries and dense granules (Figure 9A,D,F). The frequency of gold particles at the posterior region of intracellular tachyzoites was less pronounced but immunogold labeling was found in the cytoplasm near the nuclear periphery and underneath the posterior plasma membrane (Figure 9B,C,E). In contrast, immunogold TEM of extracellular tachyzoites (Figure 10) confirmed the almost exclusive and pronounced apical location of NcCDPK1, with gold particles at the very tip, sometimes appearing even on the apical surface of tachyzoites, and frequently associated with micronemes. No gold particles could be detected at the posterior end of extracellular parasites.

## 3. Discussion

Previous in vitro studies have shown that BKI-1294 treatment inhibited host cell invasion but did not act parasiticidal. As a response to the treatment, intracellular tachyzoites differentiated into MNCs [10,11]. Despite this interference, MNCs remained viable and metabolically active for extended periods of time. Even after continuous treatments with 5 µm BKI-1294 for 20 days, MNCs had re-differentiated into tachyzoites that then resumed proliferation [11]. 

The ultrastructure of MNCs was investigated by TEM, which demonstrated that the entire MNC surface was formed by a triple-layered plasma membrane, as in tachyzoites. The membrane separating the individual zoites was much thinner compared to the outer plasma membrane surrounding the MNCs. In addition, TEM at higher magnification showed that the inner membrane was branching off from the triple-layered plasmalemma and surrounding the entire newly formed zoites, which themselves were not able to disconnect from the complex and remained trapped inside the host cell. Thus, BKI-1294 interferes with the completion of the lytic cycle by inhibiting the disjunction of individual zoites from each other. This leads to the formation of MNCs that form a schizont-like, multinucleated and intracellular organism, which contains numerous apical complexes with conoid, rhoptries, micronemes, dense granules, and mitochondria. 

Which developmental steps occur during MNC differentiation and de-differentiation? During the first four days of continuous drug treatment, NcSAG1, the major surface antigen-specific for tachyzoites [18,19], is clearly visible on the plasma membrane surrounding the MNCs. However, from day six onwards, NcSAG1 moves from the outer membrane into the interior of the complex. We could not ascertain whether NcSAG1 became associated with specific structural entities within the MNC. Although NcSAG1 was not present on the MNC surface anymore, the triple-layered plasma membrane was still present and intact after nine days of drug treatment as shown by TEM (see also [10]). Thus, it is conceivable that (an)other member(s) of the surface antigen glycoprotein-related sequences (SRS) superfamily could potentially be expressed on the surface of these complexes ([20]). 

How do MNCs re-differentiate into infective tachyzoites? Following the release of drug pressure after four days of treatment, re-differentiation into novel infective tachyzoites takes up to 10 additional days. In the first phase of two to four days, NcSAG1 surface labeling of these MNCs adopted a less clearly delineated appearance. Some NcSAG1 deposits were shed from the surface and were detected as a diffuse staining outside of the MNC but adjacent to the MNC. Staining with anti-IMC1 antibody, reminiscent for newly formed zoites, appeared closer to the MNC surface from day two pdr onwards. During subsequent timepoints, NcSAG1 staining largely disappeared from the surface but became more prominent in the MNC interior. However, we could not determine whether this interior staining was associated with any distinct structural entities within the MNC. In addition, IMC1-positive parasites were frequently detected to be associated with the MNC surface already on day four pdr. From days 7–10 onwards, the number of newly formed zoites stained by anti-IMC1 antibody increased markedly, and such zoites were increasingly found in the periphery of the MNCs, seemingly heading outwards. Newly formed PVs containing obviously proliferating tachyzoites could be detected from day 10 pdr onwards. This indicates that the block of disjunction was finally alleviated. Zoites had emerged from their MNCs, and infected neighboring cells and converted into NcSAG1-positive tachyzoites. We could not determine whether these emerging zoites already expressed NcSAG1, or an alternative SRS protein. However, we did not detect emerging zoites that were clearly stained with both antibodies during this final stage of re-differentiation. 

The steps during MNC formation and re-differentiation into tachyzoites after drug removal are schematically depicted in Figure 11. The classical steps of the lytic cycle, including initiation, elongation and emergence of daughter zoites appear to occur continuously in these MNCs, but the daughter zoites fail to undergo disjunction. Consequently, a schizont-like stage appears, which exhibits an outer NcSAG1-containing plasma membrane and is able to survive for prolonged periods of time. Upon removal of the drug, NcSAG1 gradually disappeared from the MNC surface. The temporal block in completing cytokinesis was alleviated, which eventually leads to the emergence and subsequent continuous proliferation of tachyzoites, that either do, or do not, express NcSAG1.

NcCDPK1, the molecular target of BKI-1294 and related compounds, exhibits a dispersed and homogenous cytoplasmic, extranuclear localization in intracellular tachyzoites, as shown by IF and immunogold-on-section labeling. The same pattern occurred in MNCs formed after three days of BKI-1294 treatment, and subsequently, NcCDPK1 staining intensity was greatly reduced. These findings are not in agreement with previous studies with GFP- and HA-tagged TgCDPK1 localized in the cytoplasm as well as in the nucleus of *T. gondii* tachyzoites [7]. The different results could be explained by the overexpression of tagged proteins. Moreover, we have evidence that, under these experimental conditions, the apical shift seen in *N. caninum* tachyzoites did not occur in *T. gondii* tachyzoites of the RH and ME49 strains, and also not in the closely related *B. besnoiti* (data not shown).

A relocation of NcCDPK1 is only seen in tachyzoites that are isolated at elevated temperatures (21 °C). Since DTT treatment has the same effect, it is conceivable that a sudden increase of intracellular calcium triggers this relocation [22]. The relocated NcCDPK1 is localized in the sub-membranous compartment of the apical complex, near the conoid, and within some, but not all, micronemes, as well as on the tachyzoite surface, but not in rhoptries nor dense granules. The localization in the micronemes implies that NcCDPK1 could be a part of the secreted fraction of *N. caninum* tachyzoites. However, secretion assays showed that only small amounts of the total cellular NcCDPK1 are found in the extracellular medium. Therefore, NcCDPK1 cannot be considered as a major component of the secreted fraction. Previous studies have shown that another CDPK, TgCDPK3, is also localized to the inner side of the plasma membrane. TgCDPK3 is the orthologue of *Plasmodium spp.* CDPK1, and regulates Ca^2+^ ionophore- and DTT-induced host cell egress, but neither motility nor invasion [23]. In the chicken parasite *Eimeria tenella*, CDPK3, involved in host cell invasion, exhibits a dispersed localization during the first schizogony and intense specific staining at the apical end of the sporozoites after early infection of host cells [24].

In conclusion, our results suggest that MNCs formed as a response to treatment with BKI-1294 constitute an intracellular stage that enables survival and limited proliferation under adverse conditions, similar to the tissue cyst encapsulated bradyzoite stage. Previous findings showed BKI-1294 treatment triggered the increased expression of both, SAG1 and BAG1 transcript and protein expression [11] (MNC formation is also observed with *T. gondii* [11], *B. besnoiti* [25], and *S. neurona* [16], and this not only upon treatment with BKI-1294, but also upon treatment with other BKIs. MNC formation may also occur as a response to other, non-parasiticidal drugs and could, therefore, constitute an important strategy in drug adaptation or resistance. Therefore, it may be worthwhile to screen for drug combinations that do not lead to MNC formation, and regrowth in culture, to avoid the possibility of residual infection in vivo. However, it is equally possible that the MNCs are more susceptible to host innate and acquired immunity, and may be eliminated during drug treatment. In vivo experiments are required to determine if MNC formation becomes a persistence issue after drug therapy. 

## 4. Materials and Methods 

### 4.1. Tissue Culture Media, Biochemicals, and Drugs

If not stated otherwise, all tissue culture media were purchased from Gibco-BRL (Zürich, Switzerland), and biochemicals were from Sigma (St. Louis, MO, USA). Bumped kinase inhibitor-1294 was provided by the Center for Emerging and Reemerging Infectious Diseases (CERID), Division of Allergy and Infectious Diseases, Department of Medicine, University of Washington (Seattle, WA, USA), and was stored at room temperature. For in vitro studies, 20 mm stock solutions were prepared in dimethyl sulfoxide (DMSO) and they were stored at −20 °C.

### 4.2. Host Cell Cultivation and Parasite Maintenance

Human foreskin fibroblasts (HFF; ATCC^®^ SCRC-1041^™^) and BALB/c dermal fibroblasts (ATCC^®^ CCL-163) were maintained in Dulbecco’s modified Eagle medium (DMEM), and the monkey kidney cell line Marc-145 (ATCC^®^ CRL-12231) was cultured in DMEM without sodium pyruvate and HEPES. Media contained phenol red and were supplemented with 10% heat-inactivated and sterile filtered fetal calf serum (FCS), 100 U of penicillin/mL, and 100 µg streptomycin/mL. The *N. caninum*-Spain7 (Nc-Spain7) isolate was maintained by infecting semi-confluent HFF or Marc-145 monolayers and cultivation at 37 °C and 5% CO_2_, with passages once or twice per week. Tachyzoites were separated from host cells by scraping the cell material from culture flasks when they were still largely intracellular (>90% of undisrupted PVs), and infected cells were repeatedly passed through a 25-gauge needle at 4 °C [26]. In some experiments, parasite-host cell separation was performed at 21 °C, either in the presence or absence of 5 µm BKI-1294.

### 4.3. Analysis of N. caninum Tachyzoite Triton-X-100 and Secreted Fractions by Western Blots

Purified *N. caninum* tachyzoites were subjected to Triton X-100 extraction as previously described [27], and excretory/secretory fractions were prepared according to a previously published method [28]. Fractions corresponding to the same number of tachyzoites, were separated by sodium dodecyl sulfate-polyacrylamide gel electrophoresis (SDS-PAGE) and immunoblots were prepared by transfer onto nitrocellulose. Non-specific binding sites were blocked in 3% bovine serum albumin (BSA) in TBS-Tween (20 mm Tris–HCl, 150 mm NaCl, 0.3% Tween 20) for 2 h at room temperature. Blots were labeled with rabbit anti-TgCDPK1 antiserum diluted 1:5000, mouse monoclonal anti-tubulin (clone 5–2-2; Sigma) at 1:2000, and mouse monoclonal anti-NcSAG1 (1:5000; [11]), all diluted in TBS-Tween/0.3% BSA overnight at 4 °C. Bound antibodies were visualized using goat anti-rabbit-alkaline phosphatase conjugates (Promega) according to the instructions provided by the manufacturer.

### 4.4. Immunofluorescence (IF) Labeling of Intracellular Tachyzoites

For staining of intracellular parasites, freshly split HFF were seeded onto glass coverslips placed into 24-well tissue culture plates at 2 × 10^4^ tachyzoites/well in 1 mL DMEM cell culture medium/well [11]. After three days, a semi-confluent monolayer had formed, which was then infected with 2 × 10^5^ freshly purified Nc-Spain7 tachyzoites at 37 °C/5% CO_2_. At 4 h post-infection (p.i.), the medium and non-adherent parasites were aspirated and replaced by fresh culture medium, either supplemented with BKI-1294 leading to MNC formation, or the corresponding amount of DMSO, and samples were maintained at 37 °C/5% CO_2_ for different time periods as indicated in the text. In some experiments, drug pressure was removed after four days and parasite cultures were allowed to recover during extended periods of time thereafter. For fixation, coverslips were rinsed twice in PBS, and were then placed into 3% paraformaldehyde in PBS for 10 min at RT, followed by immersion in a 1:1 mixture of methanol/acetone for 20 min at −20 °C [29]. Subsequently, the samples were rehydrated in PBS at RT for 3 min, and were stored overnight at 4 °C in PBS/3% BSA to block unspecific binding sites. They were then incubated for 25 min with the primary antibodies diluted in PBS/0.3% BSA. The polyclonal anti-TgCDPK1 antibody was used at a dilution of 1:500, the mouse monoclonal antibody directed against NcSAG1 at 1:2000 [11], rabbit anti-IMC1 antiserum (kindly provided by Prof. Dominique Soldati, University of Geneva) was applied at 1:500, monoclonal mouse anti-tubulin (clone B-5–1-2) at 1:300, and monoclonal anti-NcMIC2 (kindly provided by Dr. Gereon Schares, FLI) was used at 1:4. Following incubations in primary antibodies, specimens were washed in PBS and were incubated with secondary antibodies (anti-mouse Tetramethyl-rhodamine-isothiocyanate (TRITC) or anti-rabbit Fluorescein-isothiocyanate (FITC)) both at a dilution of 1:300. For double staining, the layers were applied sequentially, 30 min each. Following antibody staining the samples were mounted in H-1200 Vectashield mounting medium with DAPI (4′,6-diamidino-2-phenylindole; Vector Laboratories, Inc. Burlingame, CA, USA), and viewed on a Nikon E80i fluorescence microscope. Processing of images was performed using the Openlab 5.5.2 software (Improvision, PerkinElmer, Waltham, MA, USA).

### 4.5. IF-labeling of Extracellular Tachyzoites 

T25 cell culture flasks, containing semi-confluent HFF monolayers were infected with 1.2 × 10^6^ Nc-Spain7 tachyzoites and cultured for 72 h at 37 °C/5% CO_2_. Isolation of parasites was carried out using different protocols as follows. (A) Standard 4 °C protocol: infected HFF monolayers were scraped and passed through a syringe with a G25 needle at 4 °C to liberate the tachyzoites; (B) 21 °C protocol: same procedure as in A, but all steps were undertaken at 21 °C and in some experiments, cultures were treated with 5 µm BKI-1294 for 30 min, prior to scraping and during the liberation of tachyzoites; (C) PFA-pre-fixation procedure: infected HFF monolayers were exposed shortly to 3% PFA for 5 min prior to scraping, and braking up cells by syringe passage, all carried out at 21 °C in the presence of PBS/3% PFA. Subsequent to parasite isolation, parasites, were centrifuged at 1000 × *g* at 21 °C (samples B and C) and 4 °C (sample A). The pellets were carefully resuspended in 3% PFA in PBS and applied to poly-L-lysine coated glass coverslips for 25 min at 21 °C, followed by centrifugation for 5 min at 1200 rpm and 21 °C. Tachyzoites on coverslips were processed for IF and viewed as described above.

### 4.6. Transmission Electron Microscopy (TEM)

To visualize the ultrastructure of MNCs, semi-confluent HFF were grown in T25 tissue culture flasks were infected with Nc-Spain7. At 4 h post-infection the medium containing non-invaded parasites was aspirated and fresh culture medium containing 5 µm BKI-1294 was added. Control cultures received the corresponding amount of DMSO. Cultures were maintained at 37 °C/5%CO_2_ for two, four, and nine days with daily supplementation of fresh medium plus/minus drug. Subsequently, infected monolayers were washed once with 100 mm sodium cacodylate buffer (pH 7.3) and fixed in cacodylate buffer containing 2.5% glutaraldehyde for 10 min. Monolayers were detached by careful scraping and the suspension was centrifuged for 10 min at 1200 rpm at 21 °C. The supernatant was removed and a fresh fixation solution was added. After incubation at 4 °C overnight, cells were post-fixed in 2% OsO_4_, dehydrated in a graded series of ethanol, and embedded in Epon 820 epoxy resin for three days with daily resin changes [11]. Specimens were polymerized at 65 °C overnight, ultrathin sections were cut on a Reichert and Jung ultramicrotome (Reichert and Jung, Vienna) and were placed on 300 mesh formvar-carbon coated nickel grids (Plano GmbH, Marburg, Germany). They were stained in Uranyless^™^ and lead citrate (EMS, Hatfield, PA, USA) as described [30], and specimens were inspected on a CM12 TEM operating at 80 kV. 

For immunogold-TEM, infected cultures at the onset of host cell lysis (partially intracellular and extracellular tachyzoites) were fixed in 3% paraformaldehyde/0.05% glutaraldehyde for 2 h at 21 °C, then washed in PBS. Samples were stepwise dehydrated in ice-cold ethanol and embedded in LR-white acrylic resin at −15 °C for three days with daily resin changes [29]. Sections were cut by ultramicrotomy as above, and on-grid section labeling was performed. Following blocking of unspecific binding sites by incubating sections in PBS/50 mm glycine/3% BSA for 1 h at 21 °C, the rabbit anti-TgCDPK1 antiserum, or the corresponding rabbit pre-immune serum, was used at a dilution of 1:250 in PBS containing 0.3% BSA for 1 h at 21 °C. After washing in five changes of PBS (2 min each), goat anti-rabbit secondary antibodies conjugated to 10 nm diameter gold particles (Aurion, Wageningen, NL) were applied at a dilution of 1:5 in PBS/0·3% BSA for 1 h. After extensive washing in PBS, grids were air-dried and stained with Uranyless^™^ and lead citrate then viewed on a CM12 TEM operating at 80 kV. 

## Figures and Tables

**Figure 1 pathogens-09-00382-f001:**
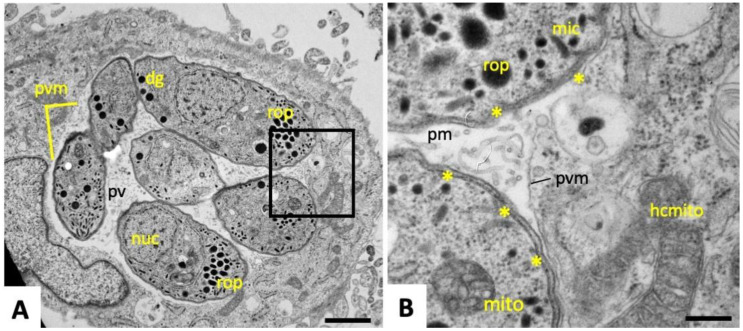
TEM of proliferating non-drug-treated N. caninum tachyzoites. The boxed area in (**A**) is enlarged in (**B**); pv = parasitophorous vacuole; pvm = parasitophorous vacuole membrane; pm = parasitophorous vacuole matrix; dg = dense granules; rop = rhoptries; mic = micronemes; mito = mitochondrion; hcmito = host cell mitochondrion; * indicate the triple-layered outer membrane of individual tachyzoites. Bars in (**A**) = 0.9 µm, in (**B**) = 0.2 µm.

**Figure 2 pathogens-09-00382-f002:**
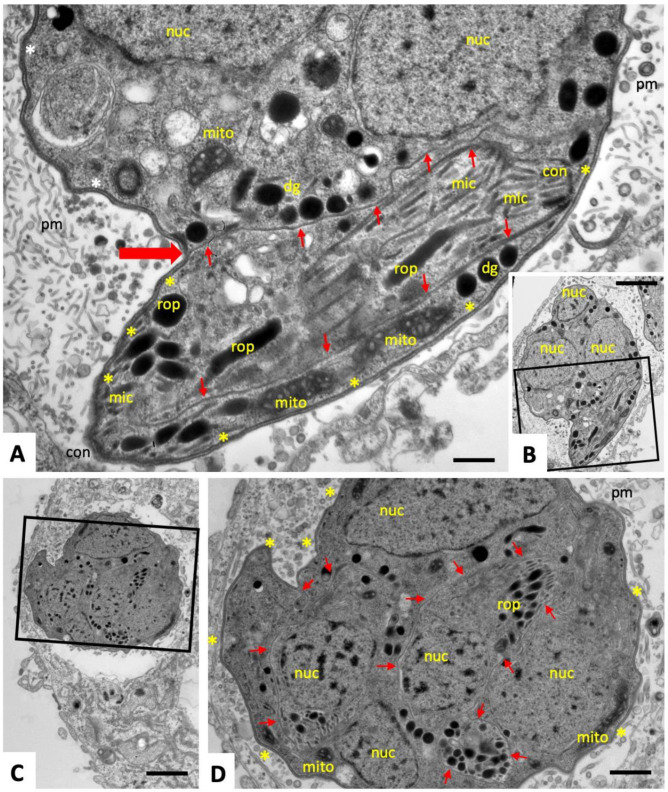
TEM of MNCs after two days (**A**,**B**) and four days (**C**,**D**) of BKI-1294 treatment. The boxed areas in (**B**) and (**C**) are enlarged in (**A**) and (**D**), respectively; pm = parasitophorous vacuole matrix; nuc = nucleus; dg = dense granules; rop = rhoptries; mic = micronemes; mito = mitochondrion; con = conoid; small red arrows point toward the IMC that separates individual zoites; the large red arrow points toward a membrane branching point; * indicate the triple-layered outer membrane of the MNC in (**A**). Bars in (**A**) = 0.2 µm, (**B**) = 1.0 µm, (**C**) = 1.8 µm, (**D**) = 0.8 µm.

**Figure 3 pathogens-09-00382-f003:**
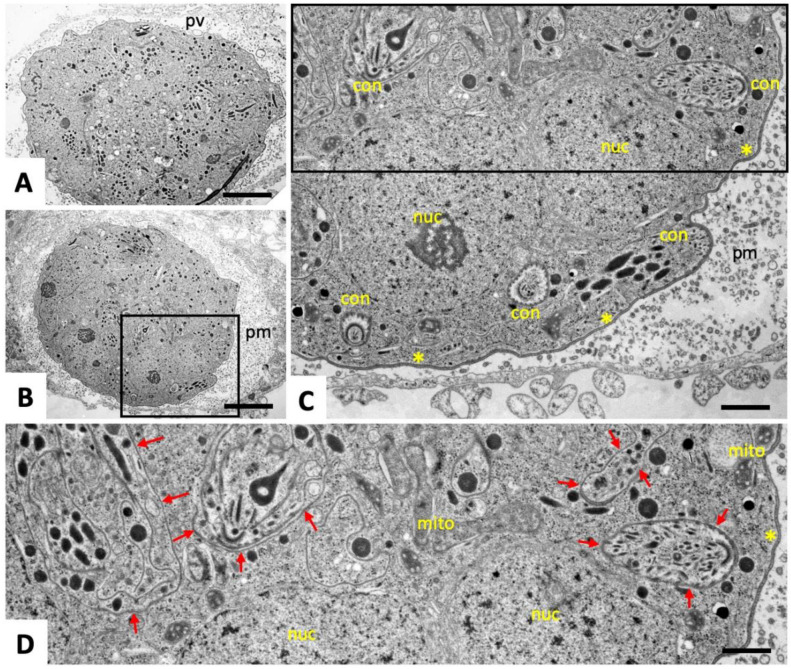
TEM of MNCs after nine days of continuous BKI-1294 treatment. The boxed areas in (**B**) and (**C**) are enlarged in (**C**), and (**D**), respectively; pv = parasitophorous vacuole; pm = parasitophorous vacuole matrix; nuc = nucleus; mito = mitochondrion; con = conoid; small red arrows point toward the IMC that separates individual zoites; * indicate the triple-layered outer membrane of the MNC. Bars in (**A**) and (**B**) = 2.8 µm, (**C**) = 0.8 µm, (**D**) = 0.6 µm.

**Figure 4 pathogens-09-00382-f004:**
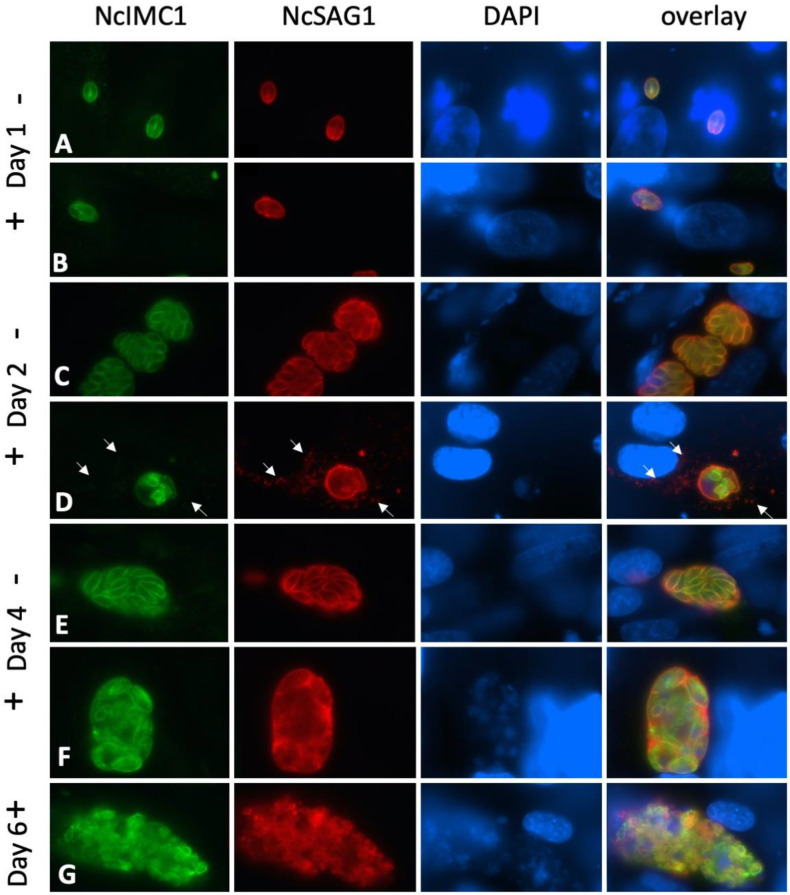
IF staining of non-treated proliferating *N. caninum* tachyzoites (**A**,**C**,**E**), and drug-treated parasites (**B**,**D**,**F**,**G**) after two, four and six days of culture in the absence (–) or presence (+) of BKI-1294. Anti-IMC1 staining is green, NcSAG1 is labeled in red, nuclei are stained with DAPI (blue). Small arrows point towards SAG1-deposits within the host cell cytoplasm.

**Figure 5 pathogens-09-00382-f005:**
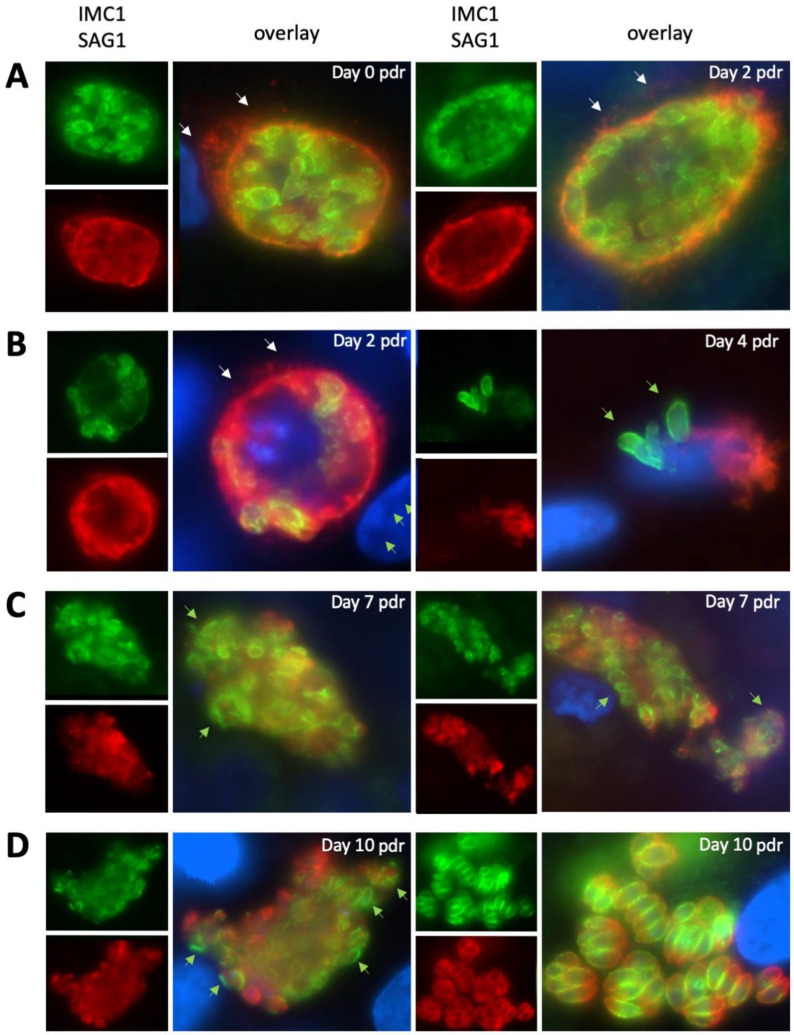
IF staining of MNCs after four days of BKI-1294 treatment followed by 10 days of culture without the drug. Specimens were fixed and processed at day zero and day two (**A**), day two and four (**B**), day seven (**C**), and day 10 post drug removal (pdr) (**D**). Anti-IMC1 staining is green, NcSAG1 is labeled in red, nuclei are stained with DAPI (blue). Small white arrows point toward SAG1-deposits within the host cell cytoplasm. Small green arrows indicate peripheral newly formed zoites stained with anti-IMC1, but not with anti-NcSAG1.

**Figure 6 pathogens-09-00382-f006:**
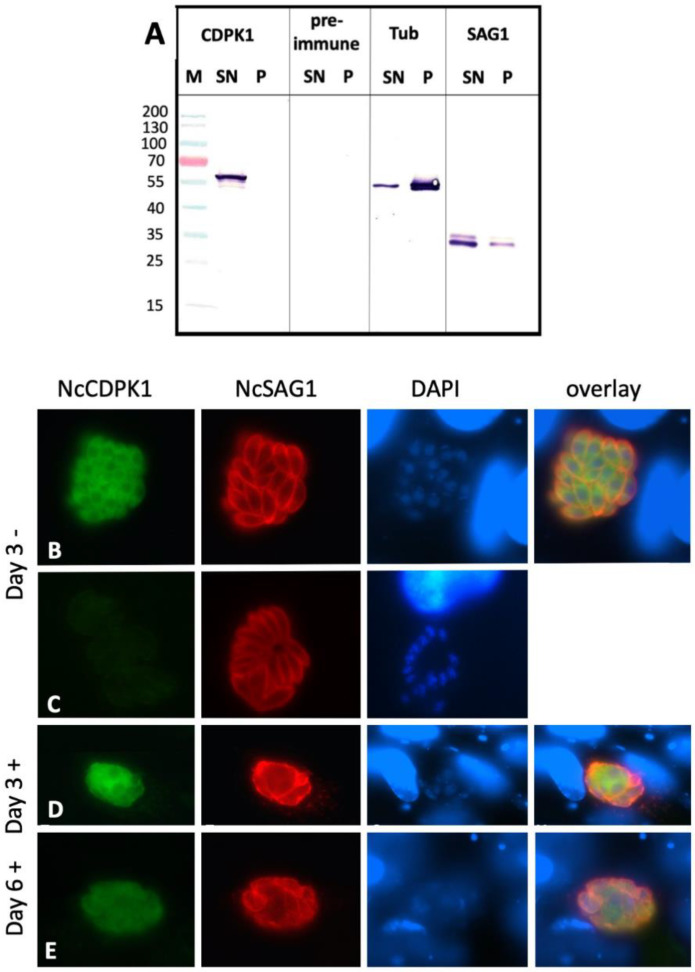
Western blots of *N. caninum* tachyzoite extracts and IF staining of MNCs. (**A**) shows *N. caninum* tachyzoites extracted in Triton-X-100, and soluble (SN) and insoluble (P) proteins were separated by SDS-PAGE and blotted onto a nitrocellulose membrane. Blots were stained with anti-TgCDPK1 (CDPK1), the corresponding rabbit pre-immune serum (pre-immune), anti-alpha-tubulin clone B-5–1-2 (Tub), or monoclonal anti-NcSAG1 antibody (SAG1). NcCDPK1-NcSAG1-double IF labeling of *N. caninum* tachyzoites cultured for three days in HFF is shown in (**B**), and (**C**) is the corresponding pre-immune serum. MNCs after three and six days of BKI-1294 treatment are shown in (**D**) and (**E**), respectively. Anti-CDPK1 staining is green, NcSAG1 is labeled in red, nuclei are stained with DAPI (blue).

**Figure 7 pathogens-09-00382-f007:**
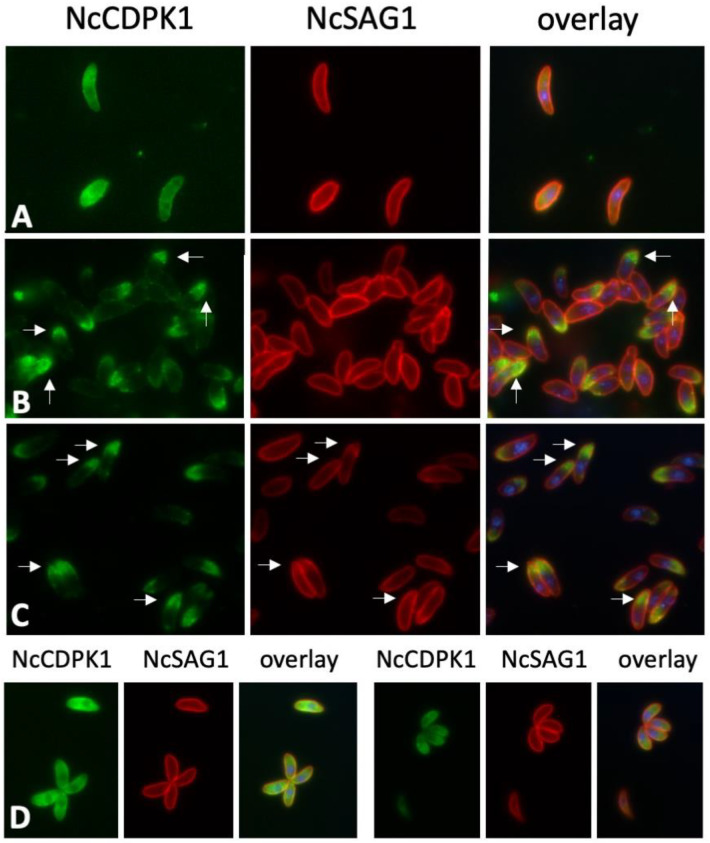
IF labeling of extracellular *N. caninum* tachyzoites. Parasites were separated from host cells either at 4 °C (**A**), at 21 °C (**B**), or at 21 °C in the presence of BKI-1294 (**C**). Panel (**D**) shows tachyzoites that were pre-fixed intracellularly with paraformaldehyde, then separated from host cells and labeled. NcCDPK1 is labeled in green, NcSAG1 is red, and DAPI in blue.

**Figure 8 pathogens-09-00382-f008:**
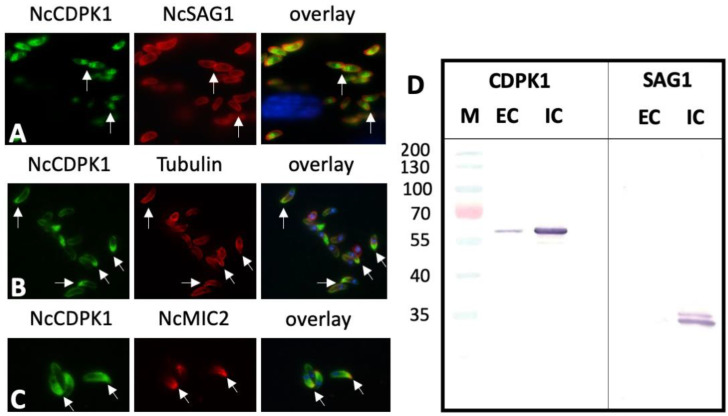
IF labeling of extracellular *N. caninum* tachyzoites and Western blots of secreted versus non-secreted fractions. (**A**) shows tachyzoites after DTT-induced egress labeled with anti-CDPK1 and anti-NcSAG1 antibodies. In (**B**) and (**C**) CDPK1-labeled parasites were double-labeled with anti-alpha-tubulin clone B-5–1-2 (Tubulin), or monoclonal anti-NcMIC2 antibody (NcMIC2). Anti-CDPK1 staining is green, NcSAG1, Tubulin, and NcMIC2 are labeled in red, nuclei are stained with DAPI (blue). (**D**) Comparative Western blot of secreted extracellular fractions (EC) versus intracellular fractions (IC) stained with anti-CDPK1 antibodies (CDPK1) and monoclonal antibody directed against NcSAG1 (SAG1).

**Figure 9 pathogens-09-00382-f009:**
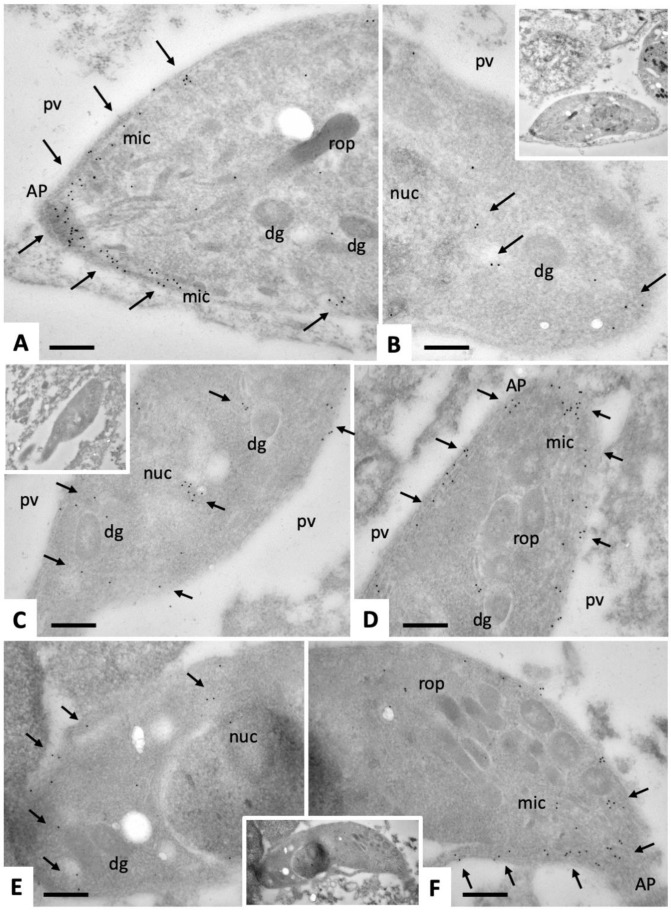
Immunogold-TEM on section labeling of intracellular *N. caninum* tachyzoites with anti-CDPK1 antibodies and secondary 10 nm gold anti-rabbit conjugates. (**A**), (**D**) and (**F**) show the apical portion of a tachyzoite. (**B**), (**C**) and (**E**) show the posterior parts, and the small inserts indicate the corresponding low magnification views. AP = apical part; pv = parasitophorous vacuole; rop = rhoptries; dg = dense granules; mic = micronemes; nuc = nucleus. Small arrows point toward gold particles indicating the presence of NcCDPK1. Bar in (**A**) and (**B**) = 0.15 µm; (**C**) and (**D**) = 0.25 µm; (**E**) and (**F**) = 0.2 µm.

**Figure 10 pathogens-09-00382-f010:**
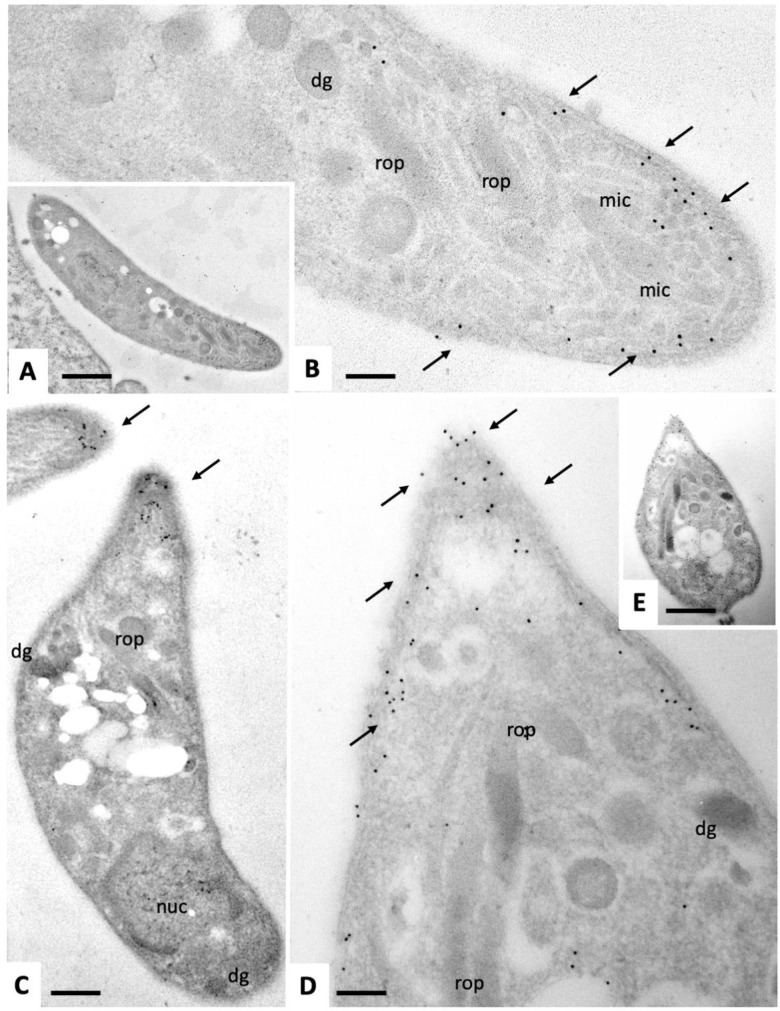
Immunogold-TEM on section labeling of extracellular *N. caninum* tachyzoites with anti-CDPK1 antibodies and secondary 10 nm gold anti-rabbit conjugates. The tachyzoite depicted in (**A**) is shown at higher magnification in (**B**), and the one in (**D**) at lower magnification in (**E**). rop = rhoptries; dg = dense granules; mic = micronemes; nuc = nucleus. Small arrows point toward gold particles indicating the presence of NcCDPK1. Bar in (**A**) = 0.8 µm; in (**B**) = 0.15 µm; in (**C**) = 0.3 µm; in (**D**) = 0.15 µm; in (**E**) = 0.5 µm.

**Figure 11 pathogens-09-00382-f011:**
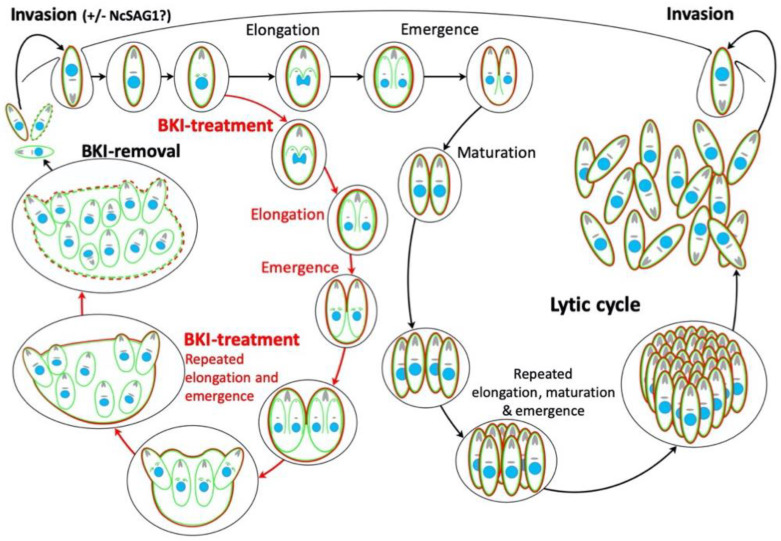
Schematic representation of the steps of the lytic cycle of *N. caninum* on the right-hand side, and the events occurring during BKI-1294 treatment and subsequent removal of the compound on the left. The different steps of the lytic cycle have been previously described by Ouologuem and Roos [21].

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
