# Peer review of "Neospora caninum: Structure and Fate of Multinucleated Complexes Induced by the Bumped Kinase Inhibitor BKI-1294"

_pathogens, 2020, doi:10.3390/pathogens9050382_

Round 1

Reviewer 1 Report

Innovative and interesting work produced by renowned authors in the field.

In my opinion, the manuscript is acceptable for publication, pending some minor adaptations.

L39 – display keywords alphabetically

L44 – Apicomplexa (uppercase)

L44 – rewrite to read as: The phylum Apicomplexa includes important pathogens of animals and/or man, including Plasmodium falciparum, Cryptosporidium parvum, Eimeria spp., Babesia spp., Theileria spp., Toxoplasma gondii and Neospora caninum.

L48 – write: T. gondii… N. caninum

L50 – delete comma after sporulation

L52 – insert comma after and,

L59 – abbreviate generic names for those above-mentioned species – is there any criteria to display these species by this order?

L76 – write: N. caninum

Explain the meaning of abbreviations at their first use, e.g. TEM and HFF

Figure 6 – overlay is missing for C

L349 – adapt: Plasmodium spp.

L358 – S. neurona

Ref. 9 – display Toxoplasma in italic type

Ref. 11 - In vitro and in vivo effects of the bumped kinase

Author Response

Thank you for your positive review. Please find below our point-by-point response:

L39 – display keywords alphabetically

Response: keywords arranged alphabetically

L44 – Apicomplexa (uppercase)

Response: done

L44 – rewrite to read as: The phylum Apicomplexa includes important pathogens of animals and/or man, including Plasmodium falciparum, Cryptosporidium parvum, Eimeria spp., Babesia spp., Theileria spp., Toxoplasma gondii and Neospora caninum.

 Response: changed as requested

L48 – write: T. gondii… N. caninum

Response: done

L50 – delete comma after sporulation

Response: done

L52 – insert comma after and,

Response: done

L59 – abbreviate generic names for those above-mentioned species – is there any criteria to display these species by this order?

Response: done. There are no specific criteria for the order.

L76 – write: N. caninum

Response: done

Explain the meaning of abbreviations at their first use, e.g. TEM and HFF

Response: done so in the first paragraph of the results section

Figure 6 – overlay is missing for C

Response: the overlay is not available. However, since there is no staining in panel A, we believe it is not explicitly necessary to show this.

L349 – adapt: Plasmodium spp.

Response: done

L358 – S. neurona

Response: done

Ref. 9 – display Toxoplasma in italic type

 Response: done

Ref. 11 - In vitro and in vivo effects of the bumped kinase

Response: corrected

Reviewer 2 Report

I have examined “Neospora caninum: Structure and fate of multinucleated complexes 

induced by the bumped kinase inhibitor BKI-1294”. I find it an excellent paper that deals with the phenotypic effects of drug treatment using a BKI on tachyzoite development. The finding of multinucleate stages in host cells but no multinucleate host cells is interesting.  This has been seen in Toxoplasma gondii treated with diclazuril but never examined in detail like the present study.  The descriptions and TEM micrographs are very good.                                                     

Author Response

Thank you for your positive review!